# Weakly Supervised Learning Approach for Implicit Aspect Extraction [†]

Aye Aye Mar *, Kiyoaki Shirai and Natthawut Kertkeidkachorn

Graduate School of Advanced Science and Technology, Japan Advanced Institute of Science and Technology,
1-1 Asahidai, Nomi 923-1292, Ishikawa, Japan; kshirai@jaist.ac.jp (K.S.); natt@jaist.ac.jp (N.K.)
* Correspondence: s1920405@jaist.ac.jp
† This article is a revised and expanded version of a paper entitled "Automatic Construction of Annotated Corpus with Implicit Aspect", which was presented at Thirteenth Edition of the Language Resources and Evaluation Conference, Marseille, France, 20–25 June 2022.

**Abstract:** Aspect-based sentiment analysis (ABSA) is a process to extract an aspect of a product from a customer review and identify its polarity. Most previous studies of ABSA focused on explicit aspects, but implicit aspects have not yet been the subject of much attention. This paper proposes a novel weakly supervised method for implicit aspect extraction, which is a task to classify a sentence into a pre-defined implicit aspect category. A dataset labeled with implicit aspects is automatically constructed from unlabeled sentences as follows. First, explicit sentences are obtained by extracting explicit aspects from unlabeled sentences, while sentences that do not contain explicit aspects are preserved as candidates of implicit sentences. Second, clustering is performed to merge the explicit and implicit sentences that share the same aspect. Third, the aspect of the explicit sentence is assigned to the implicit sentences in the same cluster as the implicit aspect label. Then, the BERT model is fine-tuned for implicit aspect extraction using the constructed dataset. The results of the experiments show that our method achieves 82% and 84% accuracy for mobile phone and PC reviews, respectively, which are 20 and 21 percentage points higher than the baseline.

**Keywords:** aspect-based sentiment analysis; aspect extraction; implicit aspect; weakly supervised learning

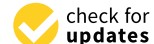



## 1. Introduction

Sentiment analysis is the task of identifying the polarity (positive or negative) of opinions written in customer reviews [1]. There are three levels of sentiment analysis: document-level, sentence-level, and aspect-level (also known as feature-level) [2]. This paper focuses on aspect-level sentiment analysis, which is also called aspect-based sentiment analysis (ABSA). It aims at inferring the sentiment of a customer at a fine-grained level by determining the polarity on each of a product's aspects, such as "price" and "battery". ABSA plays a vital role not only for customers but also for manufacturers, because it allows customers to find the strong and weak points of a product in which they are interested, while manufacturers can identify the customers' needs and expectations accurately.

In general, ABSA consists of two subtasks: aspect term extraction (ATE) and aspect polarity classification (APC). The former involves extracting aspects of a product from a sentence in a review, while the latter aims at classifying whether a customer expresses a positive, neutral, or negative opinion about each extracted aspect. Two kinds of aspects should be considered in ATE: explicit and implicit aspects. Explicit aspects are those that appear as explicit words or phrases in the review sentences, while implicit aspects are expressed implicitly, without directly mentioning the name of the aspect [3]. Some examples of these are as follows.

S1 The battery of the phone lasts many hours, so it does not need to charge frequently.

S2  I don't use it any more, as I get tired of always recharging after using just for a few hours.

Both the sentences S1 and S2 mention the same aspect of the mobile phone, the "battery". S1 contains the explicit aspect "battery" and directly expresses an opinion about it, whereas S2 implicitly expresses an opinion about the battery but without using the word "battery" itself. In the second sentence, the battery is an implicit aspect.

Although there has been a great deal of work on extracting explicit aspects, the identification of implicit aspects has not been vigorously studied [4,5]. Implicit aspects are also important in order to fully understand the opinions and sentiments of customers, since customer reviews containing implicit aspects are widespread on the internet. Zhang and Zhu showed that 30% of the reviews in their corpus contained implicit aspects [6]. Similarly, Cai et al. showed that 44% of the review sentences about laptop PCs contain implicit aspects or implicit opinions [7]. In addition, implicit review sentences are more complex than explicit ones [2]. Different people implicitly describe their sentiments about products using different kinds of linguistic expressions and writing styles, meaning that implicit aspects are more difficult to handle in ABSA than explicit ones.

The lack of a large dataset of reviews annotated with implicit aspects is one of the bottlenecks for implicit aspect extraction. Most current methods of ATE rely on supervised learning, in which an aspect extraction model is trained on a labeled dataset. The extraction of implicit aspects cannot be performed in the same way when there is no dataset labeled with implicit aspects.

The goal of this study is to develop a system of ATE for implicit aspects. A dataset labeled with implicit aspects is automatically constructed by guessing implicit aspects in unlabeled review sentences. Then, a model for implicit aspect extraction is obtained using the pre-trained language model. We demonstrate the effectiveness of our proposed method when it is applied to customer reviews in two domains: mobile phones and PCs, and present our findings on the complex nature of the implicit aspects and problems in the construction of the dataset through an error analysis. Our contributions can be summarized as follows.

- We propose a novel weakly supervised method to construct a dataset automatically labeled with implicit aspects. To the best of our knowledge, no prior work has been performed on the automatic construction of such a dataset.
- We train a model for implicit aspect extraction by fine-tuning a pre-trained language model using the above dataset coupled with existing review sentences including explicit aspects.
- We empirically evaluate the constructed dataset as well as the performance of implicit aspect extraction achieved by our proposed model.

The rest of this paper is organized as follows. Section 2 describes related work on the extraction of implicit aspects and existing labeled datasets. Section 3 introduces the proposed method and explains each component of it in detail. The results of our experiments and a discussion are presented in Section 4. Finally, Section 5 concludes the paper.

## 2. Related Work

### 2.1. Implicit Aspect Extraction

As discussed in the previous section, current methods of ABSA mainly focus on explicit aspects. However, there have been a few attempts to handle implicit aspects [4,5].

Many previous studies considered correlations between sentiment words (such as "excellent" and "bad") and aspect words. Hai et al. proposed a co-occurrence association rule mining approach for identifying implicit aspects [8]. The association rule was in the form (sentiment word → explicit aspect), indicating that the sentiment word and explicit aspect frequently co-occurred in a sentence. The rules were generated from a review corpus, then applied to identify implicit aspects of sentences that included not an explicit aspect but a sentiment word. Zeng and Li proposed a classification-based method for the identification of an implicit aspect [9]. First, pairs containing an explicit aspect and a

sentiment word were obtained using a rule-based method, where the rules were used to extract the pairs from the results of a dependency parsing of the review sentences. Then, sentences including an explicit aspect and a sentiment word were excerpted as a document collection, and were labeled with the aspect. Using this document collection as the training data, a topic–feature–centroid classifier was trained using bag-of-words features. Sun et al. proposed a context-sensitive method for implicit aspect extraction [10]. An implicit aspect in a given review sentence was identified by choosing the most related one with an explicit aspect in the context, where the correlation between the aspects was measured based on a co-occurrence matrix between aspect words and sentiment words. Bagheri et al. proposed a graph-based method for implicit aspect extraction [11]. The vertices in the graph were either explicit aspects or sentiment words, while the edges between them were weighted based on the number of their co-occurrences and the degree of the vertices in the graph. To construct the graph, explicit aspects were extracted using an iterative bootstrapping algorithm, starting with the initial seed aspects. For a given review sentence, an aspect connected to sentiment words in the sentence with highly weighted edges was extracted as an implicit aspect.

Several studies have trained models for implicit aspect extraction using a dataset labeled with implicit aspects. Hendriyana et al. proposed the sentence-level topic model (SLTM) [12]. It was a naive Bayes model that classified a review into pre-defined aspect categories. It could classify an implicit review that expressed an opinion without using explicit aspect words. Schouten and Frasincar aimed to find implicit aspects from sentences that could contain zero or more implicit aspects [13]. A training dataset was first constructed, and the co-occurrence matrix $C$ of implicit aspects and words was created from the training data. For a given sentence, scores for the implicit aspects were calculated using the matrix $C$, and the implicit aspect with the highest score was chosen. However, if the maximum score was less than a given threshold, the system judged that the sentence contained no implicit aspect.

Unlike the previous studies based on a co-occurrence matrix or correlation between sentiment words and aspects, our proposed method relies on supervised learning using a pre-trained language model that worked well for various natural language processing (NLP) tasks. In addition, instead of using a manually labeled dataset, we use a labeled dataset constructed by a weakly supervised method that requires no human effort. This enables us to develop a large dataset annotated with implicit aspects automatically.

### 2.2. Dataset of Implicit Aspects

Only explicit aspects are annotated in the most commonly used datasets for ABSA, such as Sentihood [14] and SemEval-2014 Task 4 Aspect Based Sentiment Analysis (we call it "SemEval-2014 dataset" hereafter) [15]. However, a small or pilot dataset with implicit aspects has been constructed. Hu and Liu developed a dataset for ABSA that consisted of corpora based on five product reviews: two digital cameras, a cellular phone, an MP3 player, and a DVD player [3]. Both the explicit and implicit aspects were manually annotated. However, Hu and Liu's dataset was relatively small, and the number of sentences containing implicit aspects for each of the five products was between 14 and 55. Cruz et al. extended this dataset by adding annotations of implicit aspect indicators (IAIs), which were sentiment words indicating a certain implicit aspect [16]. They selected sentences labeled with at least one implicit aspect from Hu and Liu's dataset, and then manually annotated the sentences with the IAIs. They then used the extended dataset to train a conditional random field (CRF) to extract IAIs from the review sentences.

Most methods for ABSA are based on supervised learning, which requires a labeled dataset [17]. In addition, the aspects mentioned in each review are very different for different product types or domains. To perform ABSA for various types of products, it is necessary to individually construct a labeled dataset for each domain. This is our primary motivation for the automatic construction of a large review dataset annotated with implicit aspects.

There have been a few attempts to automatically construct a dataset with explicit aspects. Giannakopoulos et al. constructed a new dataset from Amazon computer reviews [18]. First, they built a model to predict the rating of a review. During training, the model assigned weights to all sentences in a review, where higher weights indicated more important sentences. After extracting the important sentences, aspect words were automatically assigned by some heuristic rules to form a labeled dataset. Hadano et al. acquired new training data for aspect extraction based on clustering of sentences, as they assumed that similar sentences shared common aspects [19]. First, each sentence was represented by a vector consisting of content words, then a simple and fast hard-clustering tool called Bayon was used for clustering. Next, an aspect of a representative sentence in each cluster was determined by an annotator. Finally, the aspect given by the annotator was propagated to other sentences that were in the same cluster and close to the centroid of the cluster. Our method shares the same idea as this method, that is, both methods construct a dataset by clustering of review sentences. However, while Hadano's method is semi-supervised, where a human annotator determines the aspects of the sentences, our method is unsupervised, in the sense that it requires no human intervention.

## 3. Proposed Method

### 3.1. Overview

In this study, the task of implicit aspect extraction is defined as a classification problem. For a given review sentence, our system chooses a category of the implicit aspect of which the sentence implicitly expresses the reviewer's opinion. It is supposed that these categories of the implicit aspects are pre-defined. For example, the sentence "I get tired of often recharging it." is classified into the implicit aspect category "battery".

In this paper, we focus on the identification of implicit aspects for two types of products: mobile phones (or "phones" for short) and personal computers (PCs). For each, six or five categories of implicit aspects are defined, as in Table 1. The category "interface" for PCs includes any devices for the human–machine interface, such as a keyboard, track pad, mouse, and so on.

Figure 1 shows an overview of our proposed method. Since no large-scale dataset labeled with implicit aspects is publicly available, it is automatically constructed. Specifically, from a large number of unlabeled reviews and a public dataset of reviews labeled with explicit aspects, the "dataset constructor" module automatically extracts review sentences with implicit aspects to form a dataset labeled with the implicit aspects. This module is essential in this study; the details are described in Section 3.2. Next, a classifier of implicit aspects is trained from the obtained dataset. The details of these procedures are described in Section 3.3. In the inference, an implicit aspect of a test sentence is identified by the trained classifier.

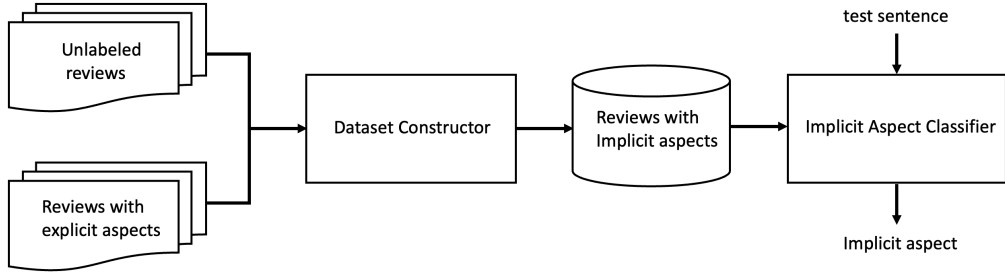

**Figure 1.** Overview of proposed method.

**Table 1.** Definition of the categories of implicit aspects.

| Product Type | Category of Implicit Aspect |
| --- | --- |
| Mobile phone | Battery, case, look, price, screen, size |
| PC | Interface, OS, price, screen, software |

*3.2. Construction of Dataset Annotated with Implicit Aspects*

Figure 2 shows how the dataset annotated with implicit aspects is constructed. Amazon reviews are used as unlabeled reviews, while the SemEval-2014 dataset is used as a set of reviews with explicit aspects. The dataset is constructed in four steps. First, explicit aspects are extracted from the Amazon reviews using an aspect extraction model trained from the SemEval-2014 dataset [15]. Second, a clustering of the sentences of the reviews is performed, where sentences that mention the same aspect, regardless of whether it is implicit or explicit, are intended to be merged into a cluster. Third, a label is determined for each cluster: the aspect of the reviews in that cluster. Finally, the sentences labeled with implicit aspects are retrieved to form the dataset. The details of these steps are presented in the following subsections.

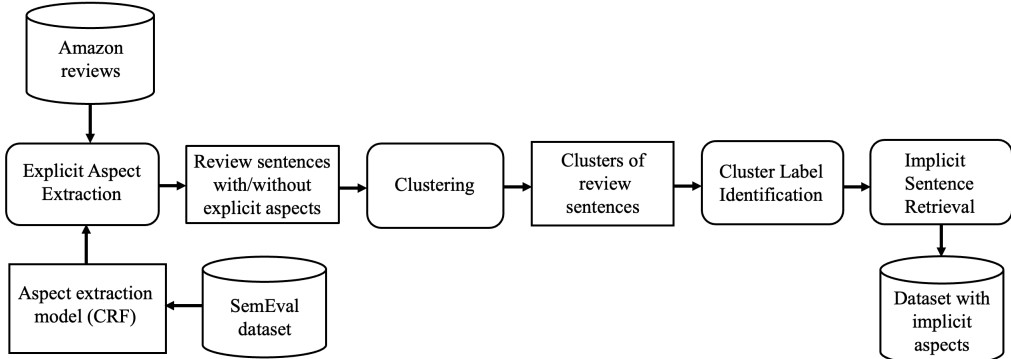

**Figure 2.** Flowchart of construction of a corpus annotated with implicit aspects.

3.2.1. Explicit Aspect Extraction

The goal of explicit aspect extraction is to extract, from unlabeled Amazon reviews, words and phrases that explicitly represent an aspect of a product. To achieve this, a model to extract explicit aspects is obtained using supervised learning. The dataset of SemEval-2014 Task 4 ABSA is used as the training data. Although there are four subtasks in Task 4 ABSA, subtask 1, "aspect term extraction", is the most appropriate for this study. All aspect terms in the review sentences are marked up in the dataset. The task organizers provide two domain-specific datasets: one for laptops, and the other for restaurants. Each consists of around 3000 reviews. This is one of the largest publicly available datasets for ABSA. In the present study, the laptop dataset is used for extracting explicit aspects for the PC domain. It is also used for the mobile phone domain, since there is no available dataset of ABSA for the phone domain. Note that the disagreement of the domains, between the training and test data, may decrease the performance of the explicit aspect extraction. A possible solution is to apply a domain adaptation technique that enables us to train an accurate classification model from training data of a different domain.

Conditional random field (CRF) [20] is used as a model for explicit aspect extraction. CRF can be used for sequential labeling, that is, CRF accepts a review sentence (sequence of words) as input and identifies a label for each word as output. The output labels follow the IOB format where B, I, and, O stand for the beginning, inside, and outside of an aspect word or phrase. It is known that CRF performs relatively well even when it is trained on a small training dataset. We used the sklearn-crfsuite library with the default settings. The features used to train CRF are a word, its part of speech (POS), its shape (e.g., whether

all characters are upper case), and so on. These features are extracted from the previous, current, and succeeding words.

By applying the trained CRF model, review sentences annotated with explicit aspects are extracted from the unlabeled review dataset. In addition, review sentences from which no explicit aspect has been extracted are also retrieved. These sentences might include no aspect, but sometimes could include an implicit aspect. In other words, the sentences without an explicit aspect can be sentences potentially including implicit aspects. As a result, a set of review sentences with and without explicit aspects is obtained at the explicit-aspect-extraction step. Table 2 shows examples of extracted sentences (in this paper, outputs of the tokenizer are shown as example sentences. For example, "doesn't" is split into two tokens, "does" and "n't"). The second and third sentences contain the explicit aspects of "screen" and "price", respectively, while the other sentences include no explicit aspect.

**Table 2.** Examples of sentences obtained using explicit aspect extraction.

| Review Sentence | Aspect |
|---|---|
| It does n't click with the white piece at all, and it easily slides off | *none* |
| I wanted so much to keep this case on, but I also did n't wan na risk my phone having a giant crack on the <u>screen</u> due to a case that does n't stay on | screen |
| Or the <u>price</u>, it is neat, but I really doubt I 'm going to keep it on my phone | price |
| What a bummer | *none* |
| It is cute and light weight | *none* |

### 3.2.2. Clustering

The review sentences, either labeled with explicit aspects or unlabeled, are then merged into clusters. The goal of this clustering is to make clusters of sentences that express opinions about the same aspect.

Each review sentence is converted to sparse composite document vectors (SCDVs) [21]. SCDVs are well known as excellent vector representations of a document, and are usually better than the average of the word embeddings, which is the simplest method to obtain an embedding of a document or sentence. SCDVs are suitable for capturing semantic similarity between explicit and implicit review sentences.

We chose $k$-means as the clustering algorithm, since it is an efficient method. The distance between two vectors of sentences is measured by the Euclidean distance. In $k$-means, the parameter $k$, the number of clusters, should be determined in advance. In this study, it is not preferable to merge the sentences referring to different aspects into one cluster. In other words, the purity of the clusters should be high. Therefore, we set the number of clusters to a relatively large number so that we could create many small but accurate clusters of review sentences. Specifically, the parameter $k$ is set to 10% of the total number of review sentences.

Figure 3 shows an example of a constructed cluster. It consists of five review sentences. In general, a cluster contains two kinds of sentences. One is a sentence containing one or more explicit aspects, such as sentences (1)–(4) in Figure 3, where the explicit aspects extracted by the CRF model are indicated by being in parentheses. The other is a sentence that does not contain explicit aspects, such as sentence (5). The explicit aspects that do not correspond to any pre-defined aspect categories (and their synonyms that will be explained in Section 3.2.4), such as "hard rubber", are ignored.

(1) For the prices, was n't worth sending back & is really for those few times away from home or do n't have outlet handy & the battery gets really low anyway . (price, battery)

(2) Like it but it causes the battery to get really hot and lock the phone . (battery)

(3) I really like the design, but however the casing did not snap nicely with my phone in place . (design)

(4) took a while to get to me its really cute just hard to come off which is good and bad i guess good because its secure if you drop the phone and bad because you may have to use something to get it open to clean or switch cases in any event i like its hard rubber and design . (hard rubber, design)

(5) I would have given it one star since it really does n't hold a charge or even charge for that matter , but I decided to add another for the design of the case although the kickstand is extremely flimsy and half of the time wo n't even hold up my phone . (*none*)

**Figure 3.** Example of cluster.

### 3.2.3. Cluster Label Identification

The task of cluster label identification involves choosing the most significant aspect for a sentence cluster. This is not always obvious, since there are two or more explicit aspects in the cluster, as shown in Figure 3. Algorithm 1 shows pseudocode for this process.

---

**Algorithm 1:** Algorithm for cluster label identification

**Input** : Cluster of review sentences
**Output:** Cluster label

1　Let $s_i$ be a sentence in the cluster, and $a_i$ be the explicit aspect of $s_i$;
2　Let $Fre(a_i)$ be the frequency of $a_i$ in the cluster;
3　Let $Oc(a_i)$ be the number of occurrence of $a_i$ in the set of sentences $\{s_i\}$;
4　*label* $\leftarrow$ aspect with the maximum $Fre(a_i)$;
5　**if** *label is unique* **then**
6　　return `ReliabilityCheck(`*label*`)`
7　**else**
8　　Let $\{a_i'\}$ be the set of aspects with maximum $Fre(a_i)$;
9　　*label* $\leftarrow$ aspect with the maximum $Oc(a_i')$;
10　　**if** *label is unique* **then**
11　　　return `ReliabilityCheck(`*label*`)`
12　　**else**
13　　　return *INDETERMINABLE*
14　　**end**
15　**end**
16　**def** `ReliabilityCheck(`*label*`)`
17　　**if** $Rel(label) \geq T_r$ **then**
18　　　return *label*
19　　**else**
20　　　return *INDETERMINABLE*
21　　**end**
22　**end**

---

The basic idea of Algorithm 1 is that the most frequent aspect is chosen as the cluster label. Two kinds of frequency, $Fre(a_i)$ and $Oc(a_i)$, are considered. $Fre(a_i)$ is the number of the times an aspect $a_i$ is extracted, while $Oc(a_i)$ is the number of occurrences of the aspect in the review sentences. The explicit aspects in a cluster are compared with respect to *Fre* and *Oc* in this order, then the most frequent one is chosen as the cluster label (lines 4–11). If two or more aspects have the same *Fre* and *Oc*, the label is defined as *INDETERMINABLE* (line 13), which indicates that the cluster may be wrongly made up of sentences about different aspects.

Next, we measure the reliability of the label, which is defined as the proportion of the sentences with the explicit aspect to all sentences in the cluster, as shown in Equation (1).

$$Rel(aspect) = \frac{Fre(aspect)}{\text{number of sentences in the cluster}} \tag{1}$$

If the chosen aspect is reliable enough (the reliability is greater than or equal to a threshold $T_r$), it is chosen as the cluster label; otherwise, the label is defined as *INDETERMINABLE* (lines 16–22). The threshold $T_r$ was empirically determined for each aspect category in the experiment.

Let us explain how the label of the example cluster in Figure 3 is identified. *Fre*(design) = 2 since the explicit aspect "design" is extracted twice, and *Oc*(design) = 3 since the word "design" appears three times in the sentences. Similarly, *Fre*(battery) = 2 and *Oc*(battery) = 2. Since *Fre*(design) and *Fre*(battery) are the same, *Oc* is compared. Then, "design" is chosen because *Oc*(design) > *Oc*(battery). Finally, the reliability is assigned the measure *Rel*(design) = 2/5. If this is higher than $T_r$, "design" is chosen as the label of the cluster.

### 3.2.4. Implicit Sentence Retrieval

The last step is to collect the sentences containing implicit aspects. As explained in Section 3.1, six implicit aspects for the phone domain and five implicit aspects for the PC domain have been defined. For each implicit aspect, the cluster whose label coincides with it is chosen. To obtain more clusters, a list of synonyms of the implicit aspects is created manually, and clusters for which the label is a synonym are also chosen. For example, synonyms of the aspect "battery" are "battery case", "battery life", "power", and so on. Table 3 shows examples of the synonyms. The full list of the synonyms is shown in the second columns of Tables A1 and A2 in Appendix A. No synonyms are used for "price" and "size" of the phone domain and "price" of the PC domain. Note that the cost of constructing the list of the synonyms is much less than the manual annotation of many sentences with implicit aspect labels.

**Table 3.** Examples of synonyms.

| (a) Phone domain | |
| --- | --- |
| **Aspect** | **Synonym** |
| Battery | battery case, battery life, power |
| Case | case quality, case cover |
| Look | design, color |
| Price | — |
| Screen | screen protector, screen cover |
| Size | — |

| (b) PC domain | |
| --- | --- |
| **Aspect** | **Synonym** |
| Interface | keyboard, touchpad |
| OS | windows, windows xp |
| Price | — |
| Screen | monitor, screen size |
| Software | program, applications |

Sentences for which no explicit aspect has been extracted are then retrieved from the chosen clusters. A cluster label is attached to these retrieved sentences as their implicit aspects. In the example in Figure 3, sentence (5) is retrieved with the label "look" as its implicit aspect, since the cluster label is "design", which is a synonym of "look".

Recall that the number of the clusters in *k*-means is set to a large value (10% of total sentences). Our motivation for this is to avoid making clusters containing multiple aspects, since they cause errors in the process of the cluster label identification and retrieval of implicit sentences. Although sentences with the same aspect may be scattered to different clusters, it might not be a problem because we can retrieve sentences with the implicit aspect from each cluster.

### 3.3. Implicit Aspect Classification Using BERT

A classifier for implicit aspect identification is trained using the constructed dataset. Bidirectional Encoder Representations from Transformers (BERT) [22] is chosen as our classification model, since it has achieved outstanding performance for many NLP tasks. The bert-base-uncased (https://huggingface.co/bert-base-uncased (accessed on 11 November 2023)) is chosen as the pre-trained BERT model. Then, it is fine-tuned using the dataset of implicit aspect sentences we have constructed.

In addition, the SemEval-2014 dataset is also used for fine-tuning. Although it contains not implicit but explicit aspects, as we discussed, the linguistic expressions in sentences with explicit aspects may be similar to those of the implicit aspects. Thus, the explicit aspect sentences can also be used for fine-tuning BERT, resulting in an increase in the number of the training samples. To form the training data for the implicit aspect classifier, the review sentences with an explicit aspect that agrees with one of the pre-defined implicit aspect categories are excerpted. Similar to the retrieval of implicit aspect sentences described in Section 3.2.4, soft-matching using a list of synonyms of the implicit aspect is applied to retrieve more sentences. The synonyms for each implicit aspect category are manually and exhaustively excerpted from the SemEval-2014 dataset. The complete list of the synonyms is shown in the third columns in Tables A1 and A2 in Appendix A. In addition, a review sentence is not included in the training data when it contains two or more implicit aspect categories. For example, let us consider the sentence "this laptop is a great price and has a sleek look". Since it contains two aspects, "price" and "look", it is not included in the training data for the phone domain. Note that it is included in the training data for the PC domain, since only "price" is an implicit aspect category for this domain.

In summary, the BERT models of implicit aspect classification are trained from two different datasets: one is the dataset of the implicit aspects created by our proposed method, the other is the union of this dataset and another corpus extracted from the SemEval-2014 dataset.

## 4. Evaluation

This section reports the results of experiments to evaluate our proposed method. First, in Section 4.1, the quality and quantity of the dataset annotated with implicit aspects described in Section 3.2 is assessed. Next, in Section 4.2, the performance of the classification of the implicit aspect using the method described in Section 3.3 will be evaluated.

### 4.1. Evaluation of Dataset Annotated with Implicit Aspects

4.1.1. Experimental Setup of the Construction of the Dataset

Amazon product data [23] was used to construct the dataset. We excerpted 30,000 review sentences from the category entitled "Cell Phones" and "Accessories" for the phone domain, and 10,000 review sentences from the category entitled "computers" for the PC domain.

The CRF model for explicit aspect extraction was trained first. The laptop reviews from the SemEval-2014 dataset were used for training the CRF model for both the phone domain and the PC domain. A preliminary evaluation of the performance of the CRF model was carried out on the SemEval-2014 dataset, where 90% of the datasets was used for training and 10% for testing. The precision, recall, and F1 score for aspect extraction were 0.77, 0.64, and 0.70, respectively, which were sufficiently high for the subsequent procedures.

For each aspect category, we randomly chose 50 sentences associated with the target implicit aspect (or all of them when the number of such sentences was less than 50). The

chosen sentences were then manually judged so as to determine whether they expressed users' opinions about the implicit aspect. In our experiment, a human evaluator judged the sentence without referring its surrounding context. As an evaluation criterion, we used the accuracy, defined as the ratio of the correct review sentences containing implicit aspects to the total number of manually checked sentences.

$T_r$ is the parameter used in Algorithm 1, where the cluster label is judged as *INDE-TERMINABLE* when the ratio of the majority aspect in the cluster is lower than $T_r$. If we set $T_r$ higher, the number of the retrieved implicit sentences will be reduced, but the accuracy will be improved. In this experiment, $T_r$ was initially set to 0.1. For some implicit aspect categories, we set $T_r$ higher when the accuracy was relatively low. We argue that it is not necessary to optimize $T_r$ using validation data. Once $T_r$ is set so that the accuracy is high, we can easily increase the number of sentences with implicit aspects by using more unlabeled review sentences. However, empirical investigation of how the increase in unlabeled sentences can contribute to enlarge the dataset and to improve the performance of the implicit aspect classification should be carried out in the future.

### 4.1.2. Results of Constructing the Dataset

Tables 4 and 5 show the statistics of the constructed dataset as well as the accuracy and $T_r$ for the phone and PC domains, respectively. The third column shows the average size of the clusters and the standard deviation in the form of ave $\pm$ sd. As for the phone domain, we obtained 290 clusters for six aspects, and the average size of the clusters (the number of sentences per cluster) was between 8 and 12. Recall that each cluster consists of sentences with both explicit and implicit aspects; the numbers of each are shown in the fourth and fifth columns, respectively. As a result, from 90 to 393 implicit sentences were obtained for six aspect categories. As for the PC domain, 149 clusters were obtained in total. The number of the obtained implicit sentences was between 45 and 261 for five aspect categories.

**Table 4.** Details of constructed dataset with implicit aspects (phone domain).

| Aspect | # of Cluster | Average Size of Cluster | # of Explicit Sentences | # of Implicit Sentences | Accuracy | $T_r$ |
|---|---|---|---|---|---|---|
| Battery | 58 | $12 \pm 16$ | 274 | 393 | 0.82 | 0.1 |
| Case | 23 | $8.6 \pm 5.5$ | 104 | 94 | 0.74 | 0.1 |
| Look | 62 | $11 \pm 9.2$ | 303 | 353 | 0.58 | 0.1 |
| Price | 89 | $11 \pm 10$ | 751 | 252 | 0.78 | 0.4 |
| Screen | 31 | $8.7 \pm 11$ | 179 | 90 | 0.76 | 0.2 |
| Size | 27 | $8.4 \pm 6.2$ | 121 | 106 | 0.70 | 0.1 |

"#" means the number of clusters or sentences.

**Table 5.** Details of constructed dataset with implicit aspects (PC domain).

| Aspect | # of Clusters | Average size of Cluster | # of Explicit Sentences | # of Implicit Sentences | Accuracy | $T_r$ |
|---|---|---|---|---|---|---|
| Interface | 24 | $9.0 \pm 5.4$ | 117 | 100 | 0.62 | 0.1 |
| OS | 25 | $12 \pm 9.0$ | 145 | 163 | 0.72 | 0.1 |
| Price | 15 | $8.6 \pm 5.8$ | 84 | 45 | 0.56 | 0.3 |
| Screen | 44 | $11 \pm 7.5$ | 228 | 261 | 0.70 | 0.1 |
| Software | 41 | $10 \pm 7.0$ | 147 | 250 | 0.64 | 0.1 |

"#" means the number of clusters or sentences.

Next, we discuss the accuracy of the obtained sentences including implicit aspects. The accuracy was 0.58 or more for all aspect categories in the phone domain. The threshold $T_r$ was set higher for the "screen" and "price" categories to improve the accuracy. As for the PC domain, the accuracy of the five aspect categories was between 0.56 and 0.72. $T_r$ was set to 0.3 for the "price" category, but 0.1 (the default value) for the others.

Figure 4 shows examples of sentences with implicit aspects. The labels for the clusters are (price) and (look), and the check marks indicate the obtained implicit sentences, in which the cluster label (price or look) is annotated as the implicit aspect. Sentence (4) in cluster 1 was successfully annotated with the aspect "price", although the word "price" was not explicitly used. The sentences with check marks in cluster 2 are other good examples of the implicit aspect of "look". Note that the cluster label was identified as "look" since the majority of the explicit aspects in this cluster were identified as "design", which is a synonym for the aspect category of "look". In summary, the results indicate that our proposed method is promising in terms of automatically constructing a dataset annotated with implicit aspects.

**Cluster 1: (price)**

(1) Great price, great service from the vendor. (price, service)

(2) Cheap price for a good quality made item. (price, quality)

(3) Very pleased with this item and it was an exellent price! (price)

(4) This was such a nice small and cheap item, I had to order 2 of them, just to have one in each car. (*none*) ✓

(5) for its price, it's not too bad, with a beautiful design (price, design)

(6) good item for great price. (price)

**Cluster 2: (look)**

(1) The color options are awesome and its very portable. (color options)

(2) Very vivid colors and the car charger is an awesome bonus. (car charger)

(3) The design is amazing and the lettering is a little light but that does n't matter as long as it fit and you are satisfied with your purchase, because I was! (design)

(4) The design was ok for a cheap case, but it was not the color it should have been! ! ! ! (design)

(5) This case is beautiful and vibrant in color, it has somewhat of a grip so it does n't slip out of your hands easily. (*none*) ✓

(6) I 've always had plain solid colors , but when I saw this I thought it would look nice. (*none*) ✓

(7) ONLY THING NICE ABOUT THIS ITEM IS THE ARRAY OF COLORS. (*none*) ✓

(8) A great buy as it does not slip out of your hand and has an awesome vivid design. (*none*) ✓

(9) Nice design and color. (Nice design)

(10) like the design and color. (design)

(11) I love the leopard design and colors defiantly makes my phone unique! (leopard design)

(12) I love the design and colors. (design)

(13) The colors are vibrant , the design is unique , and the case snaps together easily and is actually hard to pry back off ( I tried! (design)

(14) I do like this Owl & case and the colors and the design is great also. (design)

**Figure 4.** Examples of sentences labeled with implicit aspects.

### 4.1.3. Error Analysis

We found some major causes of error in the process of implicit aspect identification for the phone domain. When we initially set the threshold $T_r$ to 0.1, numerous errors were found in the extraction of the implicit aspect "price". This was because "price" is a rather general concept, and frequently occurred with other aspects, such as "service", "battery", "case", or "look". For example, in cluster 1 in Figure 4, sentences (1), (2), and (5) include "price" with other aspects. In this example of a cluster, sentence (4) was correctly extracted as a sentence with this implicit aspect, but many sentences in other clusters were wrongly extracted. However, by setting $T_r$ to 0.4, the accuracy was improved to 0.78, although this was offset by a decrease in the number of extracted sentences.

"Screen" was another implicit aspect for which we found many errors. Even when sentences contained the explicit aspect "screen", they often mentioned not the screen itself

but other related concepts, such as notifications or information shown on the phone screen. However, by changing $T_r$ to 0.2, the accuracy was improved to 0.76. In addition, errors were caused by ambiguity in the meanings of words. For example, the word "look" was used both to represent the design of the mobile phone and as a verb that was almost equivalent to "seem" (as in the phrase "looks like..."). Another problem was ambiguity in the aspect itself; for example, the word "cover" was ambiguous, and could have meant "phone cover" or "screen cover".

We also found some major causes of error in the process of implicit aspect identification for the PC domain. Five percent of the manually assessed sentences labeled with the implicit aspect "software" were written about RAM. RAM might be related to the software since it enables some software and applications to run quickly. However, RAM itself is not software but hardware. In addition, 79% of the erroneous sentences for the category "interface" mentioned a port such as "USB port" and "serial port". (Note that the implicit aspect "interface" is defined as a man–machine interface such as a keyboard, mouse, or trackpad in this study. The interfaces to connect other devices (e.g., USB port, display port) are not included). They originated from one cluster that consisted of many sentences including the word "port". The label of this cluster was identified as "interface" since some sentences included both "keyboard" (a synonym of "interface") and "port", e.g., "Because your keyboard itself has 2 USB ports, you can plug your mouse and printer into your keyboard". Accidental co-occurrence of an aspect and another word (such as "keyboard" and "port") could be a cause of the incorrect assignment of the implicit aspect.

### 4.2. Evaluation of Implicit Aspect Classification

4.2.1. Experimental Setup of Implicit Aspect Classification

The performance of the proposed model for classification of implicit aspects was evaluated. First, the test dataset was constructed using the following procedures. As described in Section 4.1.1, a few sentences in the constructed dataset labeled with implicit aspects were manually evaluated. For each aspect category, 30 (or 10 when the number of extracted implicit sentences is small) sentences were randomly chosen from the sentences that were judged as correct. Thus, the test data consisted of genuine sentences with implicit aspects where the distribution of the aspect categories was relatively balanced.

Next, three datasets were constructed as follows.

$D_e$   A set of review sentences with explicit aspects. This was made from the SemEval-2014 dataset as described in Section 3.3.

$D_i$   A set of review sentences with implicit aspects. This was constructed using our proposed method as described in Section 3.2.

$D_{e+i}$ A set of both sentences with explicit and implicit aspects.

Table 6 (a) and (b) show the number of sentences in $D_e$, $D_i$, and $D_{e+i}$ as well as the test data for the phone and PC domains, respectively.

Three classifiers were obtained by fine-tuning BERT using these datasets. Hereafter, the models trained from $D_e$, $D_i$, and $D_{e+i}$ are denoted by $M_e$, $M_i$, and $M_{e+i}$, respectively. These three models were compared in this experiment, where $M_e$ is the baseline model.

**Table 6.** Statistics of the training and test data.

| (a) Phone domain | | | | |
|---|---|---|---|---|
| **Aspect** | | **Dataset** | | **Test Data** |
| | $D_e$ | $D_i$ | $D_{e+i}$ | |
| Battery | 106 | 363 | 469 | 30 |
| Case | 4 | 64 | 68 | 30 |
| Look | 21 | 323 | 344 | 30 |
| Screen | 93 | 80 | 173 | 10 |
| Size | 21 | 96 | 117 | 10 |
| Price | 82 | 222 | 304 | 30 |
| Total | 327 | 1148 | 1475 | 140 |
| (b) PC domain | | | | |
| **Aspect** | | **Dataset** | | **Test Data** |
| | $D_e$ | $D_i$ | $D_{e+i}$ | |
| Interface | 83 | 70 | 153 | 30 |
| OS | 45 | 133 | 178 | 30 |
| Price | 83 | 35 | 118 | 10 |
| Screen | 88 | 231 | 319 | 30 |
| Software | 104 | 220 | 324 | 30 |
| Total | 403 | 689 | 1092 | 130 |

When we fine-tuned BERT, the hyperparameters were optimized on the validation data. Specifically, the dataset in Table 6 was randomly split into 90% for the training data and 10% for the validation data. The optimized hyperparameters and their possible values are as follows: (1) batch size $\{8, 16, 32\}$; (2) learning rate $\{2e^{-5}, 3e^{-5}, 4e^{-5}, 5e^{-5}\}$; (3) number of epochs $\{2, 3, 4, 5, 10, 15, 20, 25, 30, 35, 40, 45, 50\}$. The best set of the hyperparameters was chosen by several criteria based on the validation data. More concretely, the criteria were checked in the following order: (1) the highest accuracy, (2) the highest macro average of the F1 score for all aspect categories, and (3) the lowest validation loss. The best hyperparameters for each dataset are presented in Table 7. The final BERT model was fined-tuned using the overall dataset (both the training and validation data) with the optimized hyperparameters.

**Table 7.** Optimized hyperparameters.

| Domain | | **Phone** | | | **PC** | |
|---|---|---|---|---|---|---|
| **Dataset** | $D_e$ | $D_i$ | $D_{e+i}$ | $D_e$ | $D_i$ | $D_{e+i}$ |
| Batch size | 8 | 8 | 8 | 8 | 8 | 8 |
| Learning rate | $3e^{-5}$ | $5e^{-5}$ | $4e^{-5}$ | $4e^{-5}$ | $5e^{-5}$ | $5e^{-5}$ |
| Number of epochs | 20 | 30 | 35 | 5 | 35 | 50 |

### 4.2.2. Results of Implicit Aspect Classification

Tables 8 and 9 show the results of the classification of implicit aspects. These tables present the precision (P), recall (R), and F1 score (F) for each aspect category, their macro average, and the accuracy (micro average). The best score among the three models is shown in bold.

Our model $M_i$ outperformed the baseline model $M_e$ for all evaluation criteria except for the recall of "battery" for the phone domain, the precision of "look" for the phone domain, and the recall of "screen" for the PC domain. In addition, large differences between $M_i$ and $M_e$ were found. The macro average of the F1 score and the accuracy of $M_i$ for the phone domain were better than those of $M_e$ by 0.21 and 0.17 points, respectively. Similarly,

an improvement of 0.18 points for the macro F1 and 0.16 points for the accuracy was found for the PC domain. Therefore, the corpus of sentences with implicit aspects which was constructed using our proposed method was an effective training dataset for implicit aspect classification. This seems reasonable, since both the test data and $D_i$ consisted of implicit sentences, while $D_e$ consisted of explicit sentences.

Comparing models $M_i$ and $M_{e+i}$, it was confirmed that the use of both the sentences with explicit and implicit aspects could further boost the performance of the classification. As for the phone domain, $M_{e+i}$ outperformed $M_i$ by 0.02 points with respect to the macro average of the F1 score and 0.03 points with respect to the accuracy. Further improvement was found in the results for the PC domain; $M_{e+i}$ was better than $M_i$ by 0.05 points in the macro F1 and 0.07 points in the accuracy. However, the F1 score of $M_{e+i}$ was worse than that of $M_i$ in two aspects: "case" and "screen" for the phone domain. Adding the sentences with explicit aspects to the dataset made by the sentences with implicit aspects did not always contribute to improving the performance.

**Table 8.** Results of implicit aspect classification (phone domain).

| Aspect | $M_e$ | | | $M_i$ | | | $M_{e+i}$ | | |
|---|---|---|---|---|---|---|---|---|---|
| | P | R | F | P | R | F | P | R | F |
| Battery | 0.68 | 0.90 | 0.77 | **0.96** | 0.87 | 0.91 | 0.91 | **0.97** | **0.94** |
| Case | 0.38 | 0.30 | 0.33 | 0.79 | **0.50** | **0.61** | **0.82** | 0.47 | 0.60 |
| Look | 0.68 | 0.57 | 0.62 | 0.65 | 0.80 | 0.72 | **0.74** | **0.87** | **0.80** |
| Price | 0.92 | 0.80 | 0.86 | **0.94** | **0.97** | **0.95** | **0.94** | **0.97** | **0.95** |
| Screen | 0.39 | 0.70 | 0.50 | **0.89** | **0.80** | **0.84** | 0.62 | **0.80** | 0.70 |
| Size | 0.43 | 0.30 | 0.35 | 0.53 | **0.90** | 0.67 | **0.75** | **0.90** | **0.82** |
| Macro avg. | 0.58 | 0.59 | 0.57 | 0.79 | 0.81 | 0.78 | **0.80** | **0.83** | **0.80** |
| Accuracy | 0.62 | | | 0.79 | | | **0.82** | | |

**Table 9.** Results of implicit aspect classification (PC domain).

| Aspect | $M_e$ | | | $M_i$ | | | $M_{e+i}$ | | |
|---|---|---|---|---|---|---|---|---|---|
| | P | R | F | P | R | F | P | R | F |
| Interface | 0.95 | 0.60 | 0.73 | **1.00** | 0.67 | 0.80 | 0.83 | **0.83** | **0.83** |
| OS | 0.82 | 0.30 | 0.44 | 0.89 | **0.83** | 0.86 | **0.93** | **0.83** | **0.88** |
| Price | 0.67 | 0.40 | 0.50 | **0.86** | **0.60** | **0.71** | **0.86** | **0.60** | **0.71** |
| Screen | 0.73 | 0.80 | 0.76 | 0.89 | 0.80 | 0.84 | **0.90** | **0.90** | **0.90** |
| Software | 0.39 | 0.80 | 0.53 | 0.52 | 0.83 | 0.64 | **0.72** | **0.87** | **0.79** |
| Macro avg. | 0.71 | 0.58 | 0.59 | 0.83 | 0.75 | 0.77 | **0.85** | **0.81** | **0.82** |
| Accuracy | 0.61 | | | 0.77 | | | **0.84** | | |

Our best model $M_{e+i}$ was always better than the baseline ($M_e$) with respect to the F1 score for all aspect categories. As for the macro average of the F1 score and the accuracy, improvements of 0.23 and 0.20 points were found. These results prove the effectiveness of our proposed method.

We discuss the reason why $M_e$, the model trained from the sentences with explicit aspects, performed poorly. One obvious reason is the lack of training samples. For example, there were only four sentences for the aspect "case" in $D_e$ for the phone domain, thus the F1 score of the model $M_e$ was low, 0.33. Another reason may be the disagreement of the domains. Recall that the dataset $D_e$ was extracted from the SemEval-2014 ABSA laptop dataset. When it was applied to the phone domain, the domains of the training and test data were different. We observed that the aspect "screen" in the phone and PC domains referred to different concepts, although it was the same aspect for electronic devices. Customers pay attention to a protector, cover, fingerprint, or swipe on the screen in the phone domain, while they focus on resolution or size of a screen in the PC domain. This might be the

reason why the F1 score for "screen" in the phone domain was low, viz., 0.50. On the other hand, in our approach, the sentences with an implicit aspect were extracted from unlabeled data of the same domain. Therefore, the problem of the domain shift can be alleviated. In addition, the increase in the number of the samples in the training data can obviously contribute to improving the performance of the classifier.

## 5. Conclusions

This paper has proposed a weakly supervised leaning method to classify a sentence in a customer review into pre-defined categories of implicit aspects. First, a dataset annotated with implicit aspects was automatically constructed. For a given unlabeled dataset consisting of sentences with explicit and implicit aspects, clustering was performed to merge sentences having the same (explicit or implicit) aspect. Then, the explicit aspect was transferred to a sentence within the same cluster as a label of an implicit aspect. Next, the constructed dataset and the existing dataset with explicit aspects were used to fine-tune the BERT model to identify an implicit aspect of a sentence. The results of an experiment on two domains (mobile phones and PCs) showed that our proposed model, trained from the weakly labeled dataset, outperformed the baseline, trained from the sentences with explicit aspects only, by a large margin. An error analysis was also carried out to reveal the major problems in the construction of the implicit-aspect-labeled dataset.

Future avenues of research continuing those of this study include the following:

- Currently, our method supposes that there is only one implicit aspect in a sentence. It is necessary to extend our method of constructing the dataset as well as implicit aspect classification to handle sentences including multiple implicit aspects. One of the possible solutions is as follows. Instead of a hard clustering, a soft clustering could be applied to allow a sentence to belong to multiple clusters. This would enable us to add multiple implicit aspects for one sentence in the dataset. Then, we could train the model for multi-class classification that could identify multiple implicit aspects in one sentence.
- The explicit aspects were automatically extracted, but some of them may have been incorrect. On the other hand, the sentences including the explicit aspects can be obtained from the existing dataset for ABSA. These sentences can be mixed with unlabeled sentences for the clustering. Such an approach may improve the performance of the clustering.
- Manual construction of the synonym lists shown in Table 3 (also Tables A1 and A2) could be replaced with an automatic synonym expansion method.
- More appropriate clustering algorithms other than *k*-means should be investigated.

**Author Contributions:** Conceptualization, A.A.M. and K.S.; methodology, A.A.M. and K.S.; software, A.A.M.; validation, A.A.M. and K.S.; formal analysis, A.A.M. and K.S.; investigation, A.A.M. and K.S.; resources, A.A.M. and K.S.; data curation, A.A.M.; writing—original draft preparation, A.A.M., K.S. and N.K.; writing—review and editing, A.A.M., K.S. and N.K.; visualization, A.A.M., K.S. and N.K.; supervision, K.S.; project administration, K.S. and A.A.M. is the first author Aye Aye Mar, K.S. is the second author Kiyoaki Shirai, and N.K. is the third author Natthawut Kertkeidkachorn. All authors have read and agreed to the published version of the manuscript.

**Funding:** This work was supported by JSPS KAKENHI Grant Number JP20K11950.

**Data Availability Statement:** Our dataset labeled with implicit aspects is available at https://github.com/ayeayemar9/corpus-labeled-with-implicit-aspects.git (accessed on 1 September 2023).

**Conflicts of Interest:** The authors declare no conflict of interest.

## Appendix A. Full List of Synonyms

Tables A1 and A2 show all synonyms of the implicit aspects for the mobile phone and PC domains, respectively. These tables include two lists of the synonyms used for different purposes. The column "synonyms for soft-matching with cluster label" shows the

synonyms to choose the clusters of review sentences including the target implicit aspects (as described in Section 3.2.4), while the column "synonyms for soft-matching with explicit aspect" shows the synonyms to extract sentences including the target explicit aspects from the SemEval-2014 Task 4 ABSA dataset (as described in Section 3.3).

**Table A1.** All synonyms of aspects for mobile phone domain.

| Aspect | Synonyms for Soft-Matching with Cluster Label | Synonyms for Soft-Matching with Explicit Aspect |
|---|---|---|
| Battery | battery case, battery life, battery percentages, battery access, battery pack, battery charge, battery charger, charger, blackberry charger brand, blackberry charger, USB charger, USB adapter, cord, USB cord, USB port, USB ports, USB plugs, car charger, USB cable, USB cables, Samsung car charger, quality charger, power, power port, power loss, power light | battery life, charger, cord, usb port, usb ports, power, power light |
| Case | case quality, case cover | case design |
| Look | design, color | design, designed, color |
| Price | — | price tag, price range, cost, costing, priced, costed, shipping, budget, value |
| Screen | screen protector, screen protectors, screen cover, screen look, precut screen protectors | screen resolutions, screen resolution, 18.4″ screen, screen graphics, looking, service center, seventeen inch screen, 17″ inch screen, 17-inch screen, 17 ince screen, 17 inch screen, resolution of the screen, screen brightness, display, monitor, surface, stock screen, screen size, acer screen, lcd, lcd screen, screen hinges,picture quality, resolution on the screen |
| Size | — | size, sized |

**Table A2.** All synonyms of aspects for PC domain.

| Aspect | Synonyms for SM [1] with Cluster Label | Synonyms for Soft-Matching with Explicit Aspect |
|---|---|---|
| Interface | keyboard, touchpad, touch pad, keyboard flex | keyboard, touchpad, touch pad, keyboard flex, Keyboard, KEYBOARD, touch pad, keys, mouse, trackpad, left mouse key, key bindings, 10-key, regular layout keyboard, right click key, touch-pad, mouse keys, island backlit keyboard, multi-touch mouse, multi-touch track pad, Apple keyboard, mouse on the pad, left button, shift key, mouse pointer, flatline keyboard |
| OS | windows xp home edition,windows media player,windows xp,windows xp pro, windows convert, operating system, os | windows xp home edition, windows media player, windows xp, windows xp pro, windows convert, operating system, os, Windows 7, operating system, operating systems, XP, Vista, Windows applications, Windows Vista, key pad, Mac OS, antivirus software, Windows XP SP2, Windows 7 Ultimate, OSX 16, Windows 7 Starter, Windows 7 Home Premium, Windows 7 Professional, Windows operating system, Windows operating systems, Windows update, Windows XP drivers, Window update, Windows, Windows Vista Home Premium, Win 7 |
| Price | — | price tag, price range, cost, costing, priced, costed, shipping, budget, value |

**Table A2.** *Cont.*

| Aspect | Synonyms for SM [1] with Cluster Label | Synonyms for Soft-Matching with Explicit Aspect |
|---|---|---|
| Screen | monitor, screen size, screen real estate, screen flickers, screen distortion | monitor, screen size, screen real estate, screen flickers, screen distortion, screen resolutions, screen resolution, screen dispaly, 18.4″ screen, screen graphics, looking, service center, seventeen inch screen, 17″ inch screen, 17-inch screen, 17 ince screen, Screen size, resolution of the screen, screen brightness, 30″ HD Monitor, display, Resolution, display, surface, stock screen, Acer screen, 17 inch screen, LCD, screen hinges, picture quality, resolution on the screen |
| Software | programs, program, isoftware, applications, software kit, software problem, itools software | programs, program, isoftware, applications, software kit, software problem, itools software, MS Applications, suite of software, system, Microsoft office for the mac, preloaded software, Microsoft office, software packages, trackpad, Software, antivirus software, Microsoft Word for Mac, MS Office, MS Office apps, Dreamweaver, Final Cut Pro 7, Photoshop, Safari, Firefox, MSN Messenger, Apple applications, music software, Office Mac applications, Word, Excel, software options, Sony Sonic Stage software, Garmin GPS software, Microsoft Office 2003, powerpoint, iMovie, iWork, Internet Explorer |

[1] soft-matching.

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
