# Peer review of "Weakly Supervised Learning Approach for Implicit Aspect Extractionâ€"

_information, doi:10.3390/info14110612_

Round 1
Reviewer 1 Report
Comments and Suggestions for Authors
The manuscript proposes a novel weakly-supervised learning approach to implicit aspect extraction from a sentence. The main contribution of the manuscript is related to the proposed approach to automatic generation of a dataset labeled with implicit aspects. In addition, a model for implicit aspect extraction is obtained using a fine-tuned pre-trained language model.
The manuscript reports on an interesting research, which is valuable in itself and thematically suited to the journal. In terms of methodological approach and presentation, there are some places in the manuscript that could be improved or additionally clarified. I suggest to the authors to consider the possibility to address the following remarks.
Remarks:
1. The authors state that each sentence is converted to a sparse composite document vectors (p. 6). It would be useful if the authors stated how they conceptualize the “distance” between two vectors (in the context of clustering).
2. Figure 3 shows an example of a cluster. However, this figure provides some explicit aspects (e.g., hard rubber) which are not included in the set of explicit aspects considered in this study (as defined in Table 1). The authors should comment whether (and if so, how) the set of explicit aspects obtained by applying the Conditional Random Field was mapped onto the set of considered aspects defined in Table 1. E.g., let us assume that, after the algorithm for cluster label identification has been applied, a cluster is assigned a particular aspect (i.e., hard rubber) that is not defined in Table 1 (or Appendix). How is this cluster considered in terms of its aspect? Please not that “hard rubber” was not included in the lists available in Appendix.
3. The parameter that represents the number of clusters in the k-means algorithm was set to 10 percent of the total number of sentences (p 6). The authors should comment on why they selected this value. My question would be the following. Let us assume that cluster c1 contains aspects a1, a2 and a3, where aspect a1 has the maximum frequency, and that cluster c2 contains the same set of aspects where aspect a2 has the maximum frequency. In addition, we keep in mind that the number of clusters (k) is significant (as provided in Tables 4 and 5). Thus, if a smaller (but still relevant) predefined number of clusters was selected, it would be possible that clusters c1 and c2 are merged in one cluster. In addition, it also would be possible that aspect a3 has the maximum frequency in the merged cluster. This would affect the overall results. It would be useful if the author discussed on this issue.
4. The k-means algorithm generates spherical shape clusters. Why do the authors believe that such clusters are suitable to represent explicit aspects?
5. The authors state that “the chosen sentences were then manually judged to determine whether they expressed users’ opinions about the implicit aspect” (p. 9). It would be useful if they also stated whether the human evaluators were considering only sentences, isolated from the surrounding context, or additional context (e.g., immediately preceding or immediately succeeding sentences) was also taken into account.
6. The authors state that it is not necessary to optimize threshold Tr using validation data and that an increase the number of sentences with implicit aspects would suffice. Although I agree that the threshold should not be optimized during the validation phase, the statement that the the performance could be improved just by increasing the dataset size is not supported in the manuscript.
7. The analysis provided in the second paragraph of page 10 (starting with “Next, we discuss the quality of ...”) should not be described as qualitative. In addition, the statement on page 2 that the authors “empirically evaluate the quality and quantity of the constructed dataset” should be reformulated (since there is no qualitative aspect in the analysis).
8. In Tables 4 and 5, in addition to the average size of cluster, the authors should provide the related standard deviations.
9. The authors state that “the disagreement of the domain between the training and test data may decrease the performance of explicit aspect extraction” (p. 6). While I agree with this statement, it deserves further discussion (e.g., how the authors mitigated this risk, etc.)
10. For the purpose of explicit aspect extraction, the size of the context was set to three (p. 5-6). Why this context size was selected?
11. Minot typo: (page 6) “two kind of sentences” → “two kinds of sentences”.
Please cf. Remark 11 in the review report.
Reviewer 2 Report
Comments and Suggestions for Authors
The paper focuses on implicit aspect-extraction using a weakly-supervised learning approach that also incorporates the BERT transformer-based model.
The topic is interesting and worth investigating. The paper includes a comprehensive enough literature review. However, the following aspects could be improved:
- the paper should better highlight the consequences of having multiple aspects mentioned in a text and how the approach should be adapted in such a scenario.
- the dataset created in the paper should be made available using a public repository such as Zenodo or Github in order to promote research reproducibility. The authors are also encouraged to share the code associated with the paper.
Round 2
Reviewer 1 Report
Comments and Suggestions for Authors
The authors have adequately addressed the remarks from my previous review report and I believe that the manuscript has been sufficiently improved to warrant publication.
Reviewer 2 Report
Comments and Suggestions for Authors
I would like to thank the authors for the changes made.